# Beta decoupling relationship between CO2 emissions by GDP, energy consumption, electricity production, value-added industries, and population in China

**Rabnawaz Khan** *

School of Internet Economics and Business, Fujian University of Technology, Fuzhou City, Fujian Province, China

* khan.rab@fjut.edu.cn

**Data Availability Statement:** All relevant data are within the manuscript and its Supporting Information files.

## Abstract

The credible sources of fossil energy efficiently are a vital cause of economic growth and considerable influence on adequate security. Whereas radiant energy positively enhances or ostensibly promotes socio-economic stability and the controlled environment. The fossil energy sources supply has become progressively stern in China and reconnoitering the beta decoupling relationships between CO2 emissions, GDP, energy consumption, electricity consumption, value-added industries, and population. The results will be favorable for illustrative the security of the valuable resources. This study adopts the extended stochastic model (STIRPAT) with Beta Decoupling Techniques (BDT). This modern technique merely employs the decoupling situation by the alpha and beta effects from 1989 to 2018 and calculates the % change in CO2 emissions by GDP growth and energy consumption. The estimated results represent negative and economic growth depends on coal and natural gas. First, CO2 emissions annually increasing cause of rapid growth, energy consumption, and electricity production, and the structural contradiction of energy remained static. Second, the Value-added industries estimated that CO2 emissions reduce by primary industries. Third, the decoupling states of CO2 emissions and population show an inverse relationship. This paper tentatively suggests China is sustainable, naturally strengthens energy output, transmutes the energy consumption structure, and advances development policies under environmental circumstances.

## 1. Introduction

Economic development sustains a close mutual relationship with primary energy consumption (PEC) of natural resources. Like global warming, developing countries attaining economic growth that assess sustainable development. Environmental issues and problems have typically grown into a worldwide concern [1–3]. As a precious relic and fossil vitality are a vital inorganic resource for industrialization and population. And promptly become the main pouring force for progression for a developed economy [4].

**Funding:** The sources of funding as for the publication is supported by the "Fujian University of Technology (CN). The funder had only a role decision to publish the accepted manuscript. The author has not received any salary from any funder. The authors received no specific funding for this work.

**Competing interests:** We have no conflicts of interest to disclose.

As a developing country, the economic growth of China bettered. The terrific pace of sustainable development also reforms economic vents that crave focusing on providing high-quality stability and a sustainable growth path. China is one of the fastest-growing economies manifest by its over 30% improvement to global economic growth. It typically prevented that it will positively enhance the vastest economy by 2030 all around the modern world. China dwelled the intensest contributor to stimulated CO2 emissions in 2009. The specific volume of observed emissions progressively increased to 9.5 Gt in 2018 [5–7]. Thus, China cautiously entered with a real picture of perennial concern at international climate change negotiations, energy consumption, industries, and ruralization. By underscoring the win-win strategies, these concerted efforts and commitment are dubious to China's economic progression. It is incredibly high leveraged at climate change [8–10]. Likely, climate change naturally resulting from anthropogenic greenhouse gas (GHG), energy consumption, industrialization, and value-added service fundamental challenges for modern societies [9]. China influences the stable environment, and its new industrial growth adequately representing a tremendous challenge for the devoted rest of the civilized world. An increasing effect on the energy resource demand and supply of the world resources market paid attention to China [5]. The massive volume of stimulated CO2 emission in China may naturally extend to economic growth because urbanization is still growing in urban communities and taking a turn of strengthening. It leads to contributes to extensive energy utilization [11, 12]. However, the tremendous growth of china results in increased CO2 emissions. China carefully considers the reciprocal relationship of CO2 emission and economic growth as a visible sign of controlling sustainable development by energy consumption, electricity production, and value-added industries. Rapid urbanization and value-added industrialization convincingly show environmental challenges [13–15].

We use the Beta decoupling method in a pair of trading techniques. It has grown to the most common research technique to determine and detect suitable pairs in data analysis [16]. We used this Beta decoupling method to examine the essential facts of CO2 emissions by GDP, energy consumption, electricity production, and value-added industries. However, the decoupling method is driven by the physical quantities associated with other weakened or do not exist.

The Beta Decoupling Technique (BDT) is using to examine the individual effects of indicators. It measures the significant influence of growth and CO2 emissions. Decoupling is an occurrence of revivals on development and growth departing from their estimated or pattern of relationships. The decoupling method takes place when two different indicators rise/fall and moving in opposing directions. For instants, one increasing and the other decreasing. BDT shows indicator pairs that have consistently move together till beta decoupling, therefore, generating a position between ±1 and ±2 [17, 18]. The stochastic method, this study analyzed energy consumption, electricity generation, value-added industries, and ruralization. Modern IPAT techniques assessing potential action, and re-conceptualize the identity of IPAT, remaining its impact(log). By disaggregating T into consumption per unit of GDP (C) and consumption per unit impact (T), it identifies like I = PACT [19]. The traditional concept of IPAT identity is analyzing the CO2 emissions, so the total emissions show that (I), population (P), per capita GDP (A), and CO2 emissions per unit of GDP (T) [20]. IPAT planned into a stochastic model, calling it STIRPAT for stochastic influence by regression on population, affluence, and technology. The stochastic model (STIRPAT) has been used to measure the effects of driving forces on a variety of environmental change by CO2 emissions [21–23].

In terms of CO2 emissions, the number of research studies eagerly examined in China. There are four distinct strands of research related to this study that can be broadly notorious as their value. The first key aspect of the literature review determined and focus on the socio-

economic driver analysis for CO2 emissions by index decomposition, production, and structure decomposition model [24–26]. The second aspect showed the related evaluation policies of CO2 emissions [27, 28]. The third aspect shows the economic growth, urbanization, and the fourth aspect literature on CO2 accounting [29–31].

They showed the decoupling concept in the prior studies. The decoupling relationship between CO2 emissions, economic growth, energy consumption, electricity production, value-added industries, and the population has also been analyzed and examined in China with different prior studies.

For instance, energy consumption and economic growth analyzed [32] decoupling on the regional bases from 2002 to 2012 and exploring the relationship between fossil energy consumption and economic development [33]. They analyzed the decoupling by secondary industry and found investment was prevalent for energy consumption from 2000 to 2016 [5]. Investigated decoupling by Topia decomposition in the power industry and economic growth in China last thirty years. The expected results showed that expensive negative and weak decoupling.

However, the prior studies that verified decoupling CO2 emissions of China from economic growth are yet distant from the eventual results and conclusions. A significant contribution of this study to the extant research is the Beta decoupling technique (BDT). (1) Mostly, existing studies sought causes for decoupling economic upheaval from CO2 emissions based on technological changes and production. (2) It used the decomposition approach because of the logarithmic mean division index (LMDI), which shows the index decomposition method. It limits the LMDI approach in case of technical efficiency and growth [34]. Therefore, it's needed to introduce a more meaningful technique to determine the significant effects of CO2 emissions by growth and energy consumption. (3) The Beta decoupling technique expanded by planned (IPAT) into a stochastic model (STIRPAT) analysis of CO2 emissions. (4) We examined the decoupling among CO2 emissions, GDP, energy consumption, electricity production, value-added industries, and China population from 1989 to 2018, while most prior studies have explored erstwhile to 1989. This research study should specify a thoughtful sound of Beta decoupling and its causes for the latest eras in China.

## 1.1 Primary energy consumption

[35–37] Illustrated that China's energy consumption increase continuously after 2014–2015, it recorded at 0.017%, and up to 2017–2018, it estimated 0.04% (show in Fig 1). The primary energy consumption is analyzing with oil, natural gas, coal, nuclear, hydroelectricity, and renewable energy consumption [38]. The World Population Review shows that China is the world's highest level of CO2 emissions. They estimated it to producing 11.535 gigatons CO2 emissions in 2019, and second the USA with 5.243 gigatons. The primary task to seek a path of coordinated development by the decoupling method among the environment and economic development.

Therefore, in this context of primary energy consumption, China's economic growth and energy consumption by taking the sanctuary of resources. The breakthrough point aim not only to provide a sufficient theoretical basis for China but also to articulate future energy development plans in the world.

## 1.2 Electricity generation

[39] Illustrated that electricity generation in China increased by 19.67% from 2017 to 2020, 41.01% in 2020–2035, and expected to rise more 13.05% in 2035–2050. In 2050, the total electricity generation in China estimates to amount to 15,324 TWh [40]. It shows that tremendous economic activities affect China, and it is the most vital contribution to increase CO2

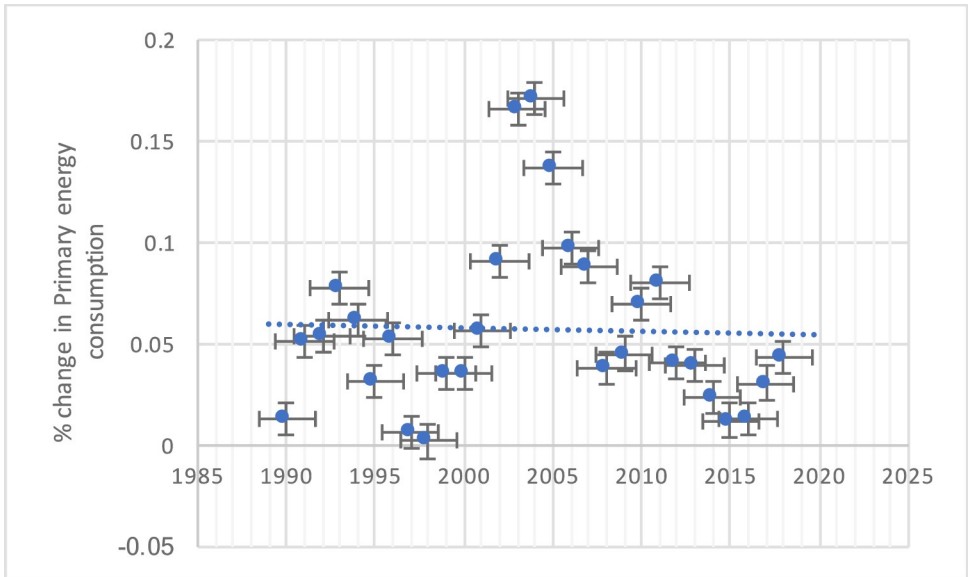

**Fig 1. % change in primary energy consumption.**

emissions from electricity generation. And efficiency influence plays the dwarf role in reducing CO2 emissions. According to Global Energy Statistical Yearbook (GESY) China's electricity production continues to grow progressively. China remains a key contributor of electricity production, with a rising thermal production, renewable, electricity production increased by 4.7% in 2019, less than half the 2000–2018 average (+10%/year). In 2019, electricity production by coal-fired decreased 3.5% offset by an increase gas-fired (+3.2%), wind (+12%), solar (+24%) and nuclear (+3.6%) production [41].

Fig 2 showed a % change in CO2 emissions because of electricity generation. A change in CO2 emission recorded the cause of gas and coal consumption from 2003 to 2005. As per the

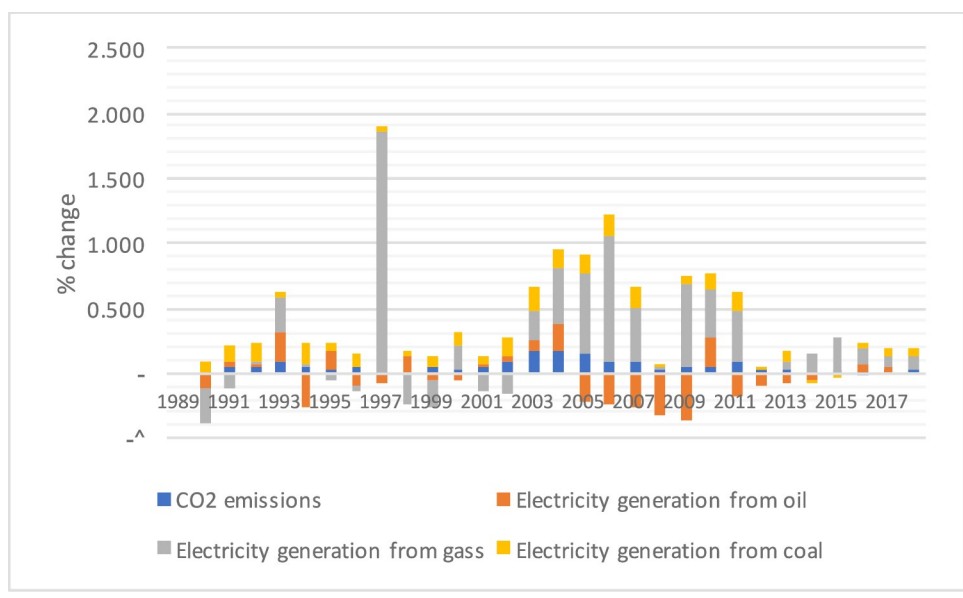

**Fig 2. % change in CO2 emissions and electricity generation production.**

excessive demand for electricity consumption, China produces electricity from other natural resources and controls the environmental effects of CO2 emissions.

## 1.3 Value-added industries

As value-added industries, China is the worlds' largest emitter of greenhouses. And it bears a grueling task of emission reduction (ER) and energy conservation (EC) from various primary, secondary, and territory industries. In 2015, the National Bureau of Statistics of China illustrated that the considerable investment in primary industries reached up to 3.8% (3,919.5 billion yuan), 6% (19,779.9 billion yuan) in secondary, and 8.4% (250,779.9 billion yuan) recorded in territory industries [42]. Additionally, they reported China 27.17% in manufacturing value added (% of GDP) during 2019. Because of the vast majority of industrial processes, CO2 arising from energy consumption and makes up greenhouse gas emissions in China. As a developing economy, China will carry out more robust strategic policies and regulations to decrease CO2 emissions in the industrial sectors. It will make efforts and strategies to control new production capacities in different industries. China has long ago established the alleged position in place of a green manufacturing system, and 171 green industrial parks and 2121 green factories build over the recent five years. China will endeavor to ultimate CO2 emissions by 2030 and producing carbon neutrality by 2060 [43]. The innovative strategies of China to reduce CO2 emissions up to 2030 from the key industries [44, 45].

However, the process of industrialization has forced China to confront the dual burden of air pollution and greenhouse gas emissions [3]. The cement is the most crucial for economic growth and surging urbanization [46]. Henceforth because of production, industrial growth is detrimental to the ecosystem, planet, and inhabitants.

Fig 3 shows a % change in CO2 emissions because of value-added industries and the highest CO2 emission recorded during 2009. China's industrial revolution has been started after 1993 and boost economic growth.

The model of this used paper implied by Beta Decoupling Techniques (BDT), which is commonly used for the % change of the indicators and shows elasticity. In this study, BDT techniques are expended for economic growth on energy consumption with the stochastic model (IPAT) and used in the discussion of the relationships among CO2 emissions, growth (GDP), primary energy consumption (PEC, OIC. NGC, COC, NUC, HCC, and RGGB),

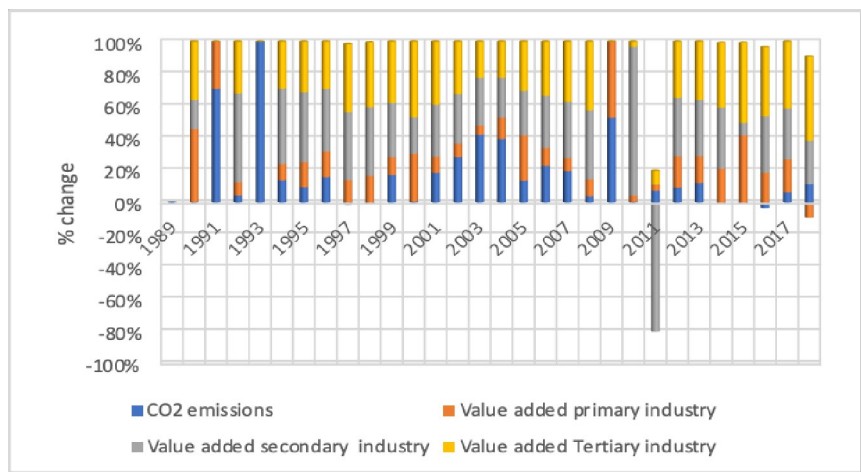

**Fig 3. Value-added industries (% change).**

electricity generation (EGO, EGS, and EGC), value-added industries (VPI, VSI, and VATI), and population (UPG and RUL). This study examines indicators effects by BDT, where IPAT is revealing four distinct clusters, and further changes of CO2 emissions have been analyzed by decoupling. However, the prior review of that research method still has paid insufficient attention to the decoupling of variant energy sources by the effects of GDP, energy consumption, electricity generation, value-added industries, and population. Disregard regarding primary energy types may lead to differentiated decoupling in the analyzed China economy during 1989–2018.

Cause of the high dependence on resources (fossil energy), China can achieve resource security and speeding up the transition to the non-fossil. The relationship between CO2 emissions and energy consumption with economic growth can not only measure mostly energy sustainability. In addition, remain a valuable practical foundation for the consciousness of China's sustainable economic growth in the upcoming years. The prior research ignores the energy consumption with value-added industries, electricity production, and the ruralization effects with GDP [47–49]. CO2 emissions in China do not lead to the business industries, and the volume extended to gigatons (GT) in 2020. They ignore value-added industries, electricity production by coal, and the ruralization effect [50].

The enormous pressure put on Chinese industries' reason for CO2 emissions. It shows to reduce CO2 emissions by a diverse way to technological implementation and proper strategies [51]. Besides, direct consumption of power and healing energies in primary and tertiary industries was lower than in secondary industries. But revealing the increase of CO2 emissions and reduced by both sides of supply and demand afterward [52].

The extant research examined the relationship between CO2 emissions, economic growth, energy consumption, electricity production, value-added industries, and the population with stochastic method from the production and development side. The stochastic (STIRPAT) method, used to calculate the CO2 emissions from 1989 to 2018.

This study discusses the BDT among between CO2 emissions change the cause of economic growth, energy consumption, electricity production, value-added industries, population, production, and consumption side in China. It compares and analyses the decoupling situation and change in % from various perspectives of STIRPAT clusters.

## 2. Methodology

### 2.1 Beta Decoupling Technique (BDT)

In this study, we select indicators according to IPAT clusters where I (CO2 emissions), P (URL and RUL), A (GDP), T (PEC, OIC. NGC, COC, NUC, HCC, RGGB, EGO, EGS, EGC, VPI, VSI, and VATI). These indicators are highly correlated and co-integrated in industries. The distribution of clusters is showing on the IPAT technique, where I indicated that CO2 emissions, P (population), A (GDP), and T (technology). Furthermore, technology is analyzing by GDP and population. This data is adjusted by covariance and variance for the computation of $\alpha$, and $\beta$.

The decoupling value based on individual change in CO2 emissions and how a 1% change affects population (P), growth (A), and technology (T). After fulfilling all conditions of clusters strategy and beta decoupling model, selected four clusters (IPAT) are shown (Table 1).

### 2.2 STIRPAT model

At for resources and environmental level of primary energy consumption, the decoupling technique used. The decoupling technique is using. It describes the relationship between

**Table 1. Variables definition and codes.**

| Variables | Code | | IPAT | Description |
|---|---|---|---|---|
| CO2 emissions | CO2 | | I | Carbon dioxide emissions (BP-data) |
| Urbanization level (10000 person) | UPG | | P | % of total population living in urban areas. |
| Ruralization (POP/1 million) | RUL | | | The total population is represented by the de facto definition of population. It is estimated on the basis of the midyear value and midyear value. |
| Gross domestic product at constant price (100 million yuan) | GDP | | A | GDP is showing the purchaser's prices, and it is the sum of gross value added by all resident. |
| Primary energy consumption (Mote) | PEC | T1 | T | Energy consumption is indicating the total energy demand in China. |
| Oil consumption Tones | OIC | T2 | | It is the amount of energy released by the burning process in different sectors. |
| Natural gas consumption | NGC | T3 | | Natural gas |
| Coal consumption | COC | T4 | | Coal consumption in industries and generation of electricity |
| Nuclear energy consumption | NUC | T5 | | Energy consumption and the generation of electricity |
| Hydroelectricity consumption | HCC | T6 | | Hydro-electricity |
| Renewable Geothermal, Biomass & others | RGGB | T7 | | Natural and organic material |
| Electricity generation from oil (Twh) | EGO | T8 | | Generation of electricity by oil, gas and coal |
| Electricity generation by gas | EGS | T9 | | |
| Electricity generation by coal | EGC | T10 | | |
| Value-added primary industries | VPI | T11 | | Value-added primary, secondary, and territory industries illustrated the Economics value that company adds to services and products before offering them to the customer, which can boost revenue and profit. |
| Value-added secondary industries | VSI | T12 | | |

Source: http://www.stats.gov.cn/tjsj/ndsj/2018/indexeh.htm

economic growth, resources, environment, value-added industries, and population. However, the environment does not change with economic growth.

This study aims at investigating the decoupling of CO2 emissions for China. This study shows economic growth, primary energy consumption, electricity generation, value-added industries, population, and decoupling of CO2 emissions with the STIRPAT model. The decoupling factors expanded by IPAT and the influence measure by GDP. Table 1 shows the indicators used for analysis, and a description and sources of data, are presented with different abbreviations. We use panel data to analyze the China data-set from 1989–2018.

The BDT of energy consumption and economic growth on the electricity generation and value-added industries are similar to those on the primary energy consumption side. The decoupling index represents the ratio of a 1% change in CO2 emissions to a 1% change in GDP. It indicated that the decoupling condition with the elastic range. In the first stage, the indicators of IPAT examine by covariance, variance, alpha, and beta. In the second stage, indicators distribute into three-level, decoupling, negative decoupling, and coupling. The critical value of positive and negative decoupling is +1 to +2 and -1 to -2.

The coupling value stated that zero and the subdivision continue with different attitudes of decoupling. The positive attitude of decoupling indicated strong, weak, and recessive conditions. And the negative decoupling is indicated that weak, strong, and expansive negative conditions. Fig 4 showing the RUL, URL, and RGGB are showing strong decoupling from 2000 to 2001 and weak decoupling with EGC from 2017 to 2018.

Since the rate of change in economic growth greater than zero, this paper contains only four decoupling states: strong-weak decoupling, expansive coupling, and expansive negative decoupling. The highest and lowest decoupling shows a % change in CO2 emissions.

The sturdiest positive attitude of VPI shows that lots of industries have an influence globally and rise the pollution level. Hence, RUL and GDP attitudes are indicated a highly negative and indirect effect on CO2 emissions by lack of technology during 2015–2020 Fig 5.

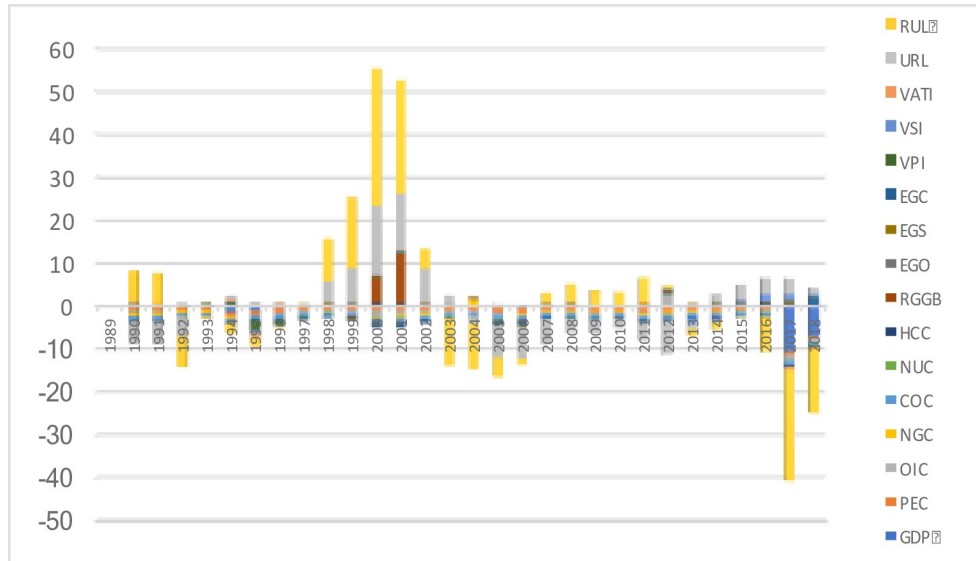

**Fig 4. Beta decoupling of indicators.**

## 2.3 Data source

China's energy consumption data come from the China Statistical Yearbook (National Bureau of Statistics, 2018). The main energy sources consume by primary energy consumption in multiple industries in China. This study selected primary-energy consumption (OIC, NGC, COC, NUC, HCC, and RGGB), electric generation (EGO, EGS, and EGC), and value-added industries (VPI, VSI, and VATI) of China.

China's population is analyzed and examine by URL and RUL. This paper concedes Beta Decoupling Techniques (BDT) [5, 53]. The BDT consisted of decoupling mechanisms by an individual change in variation (years) covariance, variance, alpha, beta, and decoupling analyzed the actual change in CO2 emissions by GDP, primary energy consumption, electricity generation, value-added industries, and population. Also, the decoupling mechanism is

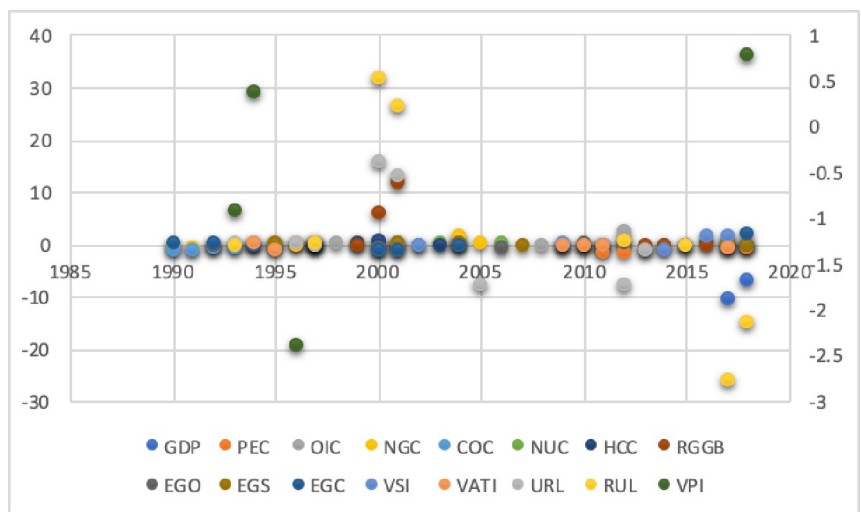

**Fig 5. Strong positive and negative decoupling.**

steadily growing, but from 1989 to 2018, it deserted. The adjustment of beta decoupling recognizes between +1 to -2, where the dependent variable (CO2 emission) will increase and decrease.

However, if the decoupling value is less than -1, we reduce CO2 emissions affects by technological skills and make proper strategies for growth and development. This paper uses GDP to indicate economic development and its influence on CO2 emissions by primary energy consumption, electricity generation, value-added industries, and population.

## 3. Results and discussion

In this analysis, the decoupling relationship between CO2 emissions by GDP, energy consumption, electricity production, value-added industries, and population used for China and all the data obtained from the National Bureau of statistics during 1989–2018.

### 3.1 Gross Development Product (GDP)

The covariance of GDP indicated that a 1% change in GDP should become a change 0.00063% change in CO2 emissions because the covariance of beta is 0.0039%. The proportion of covariance calculates by GDP and its percentage change in five years. The GDP change is 0.139% for the beta proportion and mostly changes by 0.0039% in CO2.

A 1% change in CO2 is examining at 0.081%, and the value of decoupling recorded is 0.054% with a positive attitude. The decoupling access by a 1% change in CO2 minus from the computed beta value. Therefore, we expected that the beta value changes 0.0039% by a positive attitude.

However, the reality has happened that 0.081% change in CO2, so the decoupling was 0.045, and in the next year -1.314. After the addition of both decoupling values, the expected result is -1.269. The estimated results show the record volume of CO2 emissions by % change during 2017 and 2018 with -17.501. It suggests that we should reduce the development or at least changed the industries by gross product. China has already declared that the country will reach a CO2 emissions peak within 2030 cause of vast infrastructure and development. The results showed that the volume of CO2 emissions at 12.41 Gt CO2eq (parts per gigaton; CO2 equivalent) [54].

Hence, with the support of BDT, we can compute CO2 emissions every five years. A % change in CO2 emissions is also estimated each year cause of development and technical revolution in industries. The decoupling value recorded above -2 indicates the CO2 emissions have increased due to growth and development from 1989 to 1993. The estimated decoupling of GDP record as -6.982 in the five years (2017–2021).

However, the CO2 emissions will increase in the coming years with increasing GDP. Like this, 30 years of GDP, energy consumption, generation of electricity, value-added service of industries, and population decoupling values examine in each period. The above results needed to change the strategies for CO2 emissions regarding GDP because according to the global carbon budget (2018), the global fossil CO2 emissions volume grew at a rate of 1.5% yr −1 during 2008–2017. Our estimated results (Fig 6) show decreasing effects of the BDT from 2009 to 2011 and the highest volume recorded in 2017. In this context, China's CO2 emissions increased by 3.0% yr−1 and in the next ten years to increase by 0.64% (GtC yr−1). China enters as the largest contributor to global CO2 emission in 2017, with the highest rate of 27% and the growth rate recorded as +1.5%. The per capita CO2 emissions were 1.1 (tC person−1 yr−1), and China included in the highest emitted countries list with 2.0 (tC person−1 yr−1) [55].

The New Era of China makes varied strategies to ensure health, sustainable economic development, and energy efficiency to reduce CO2 emissions from China. The new strategic

| Periods | Years | GDP | PEC | OIC | NGC | COC | NUC | HCC | RGGB | EGO | EGS | EGC | VPI | VSI | VATI | URL | RUL |
|---|---|---|---|---|---|---|---|---|---|---|---|---|---|---|---|---|---|
| | 1989 | | | | | | | | | | | | | | | | |
| 1 | 1990 | 0.079 | 1.012 | -0.467 | -0.767 | -1.287 | | 0.047 | | -0.11 | 0.035 | 0.265 | 0.128 | 0.133 | 0.142 | -4.874 | 7.473 |
| | 1991 | 0.053 | 1.065 | -0.38 | -0.742 | -1.129 | | 0.083 | | 0.032 | -0.07 | 0.136 | 0.102 | 0.11 | 0.116 | -5.243 | 7.227 |
| | 1992 | 0.03 | 0.995 | -0.094 | -0.586 | -0.664 | | 0.002 | | 0.01 | 0.062 | 0.115 | 0.336 | 0.051 | 0.071 | -4.637 | -7.099 |
| | 1993 | 0.045 | 1.003 | -0.047 | -0.65 | -0.646 | | -0.02 | | 0.028 | 0.075 | 0.143 | 0.374 | 0.069 | 0.091 | 0.146 | -0.41 |
| 2 | 1994 | -1.314 | 1.123 | 0.038 | -0.352 | -0.798 | | 0.187 | | 0.046 | 0.024 | 0.095 | 0.785 | 0.187 | 0.455 | 0.979 | -2.45 |
| | 1995 | -1.101 | 0.999 | 0.063 | 0 | -0.725 | 0.005 | 0.098 | | 0.089 | 0.015 | 0.411 | -2.38 | -0.611 | -0.961 | 0.786 | -2.064 |
| | 1996 | -0.41 | 1.079 | -0.316 | -0.04 | -0.842 | 0.063 | 0.112 | 0.041 | 0.114 | 0.057 | 0.672 | -0.908 | -0.253 | 0.294 | 0.303 | -0.517 |
| | 1997 | 0.021 | 1.036 | -0.322 | -0.029 | -0.86 | 0.268 | 0.168 | 0.057 | 0.171 | 0.036 | 0.617 | 0.021 | 0.021 | 0.021 | -0.364 | 0.08 |
| | 1998 | 0.049 | 0.934 | 0.335 | -0.227 | -0.761 | -0.18 | 0.065 | 0.081 | 0.109 | 0.064 | 0.509 | 0.049 | 0.051 | 0.049 | 4.938 | 9.938 |
| 3 | 1999 | 0.113 | 0.937 | -0.576 | 0.052 | -0.764 | -0.072 | 0.073 | -0.367 | 0.163 | 0.114 | 0.645 | 0.114 | 0.123 | 0.106 | 7.926 | 16.553 |
| | 2000 | 0.243 | 0.968 | -1.247 | -0.362 | -0.699 | 0.023 | 0.475 | 5.904 | 0.669 | 0.058 | 1.043 | 0.245 | 0.265 | 0.222 | 15.97 | 31.854 |
| | 2001 | 0.274 | 0.994 | -0.937 | -0.65 | -0.787 | 0.02 | 0.315 | 1.654 | 0.507 | 0.014 | -1.13 | 0.274 | 0.313 | 0.242 | 13.08 | 26.312 |
| | 2002 | 0.007 | -1.01 | -0.462 | -0.719 | -0.794 | 0.065 | 0.28 | 0.126 | 0.075 | 0.019 | 0.938 | 0.04 | -0.069 | 0.071 | 7.939 | 4.717 |
| | 2003 | 0.079 | 1.046 | -0.3 | -0.046 | -0.818 | 0.049 | -0.01 | 0.103 | 0.022 | 0.098 | -0.41 | 0.058 | 0.042 | 0.117 | 1.858 | -11.23 |
| 4 | 2004 | 0.144 | 1.084 | -0.391 | 1.364 | -0.831 | 0.009 | 0.066 | 0.116 | 0.097 | 0.176 | 0.008 | 0.047 | 0.129 | 0.184 | -1.607 | -10.675 |
| | 2005 | -0.23 | 1.175 | -0.622 | 0.027 | -0.901 | 0.208 | 0.199 | 0.035 | -0.21 | 0.064 | 0.664 | -0.199 | 0.129 | 0.094 | -7.637 | -4.547 |
| | 2006 | -0.437 | 1.123 | -0.099 | -0.376 | -0.764 | 0.248 | 0.15 | -0.051 | 0.733 | 0.038 | 0.609 | -0.275 | -0.178 | -0.186 | -7.215 | -1.497 |
| | 2007 | 0.045 | 1.094 | 0.168 | -0.354 | -0.79 | 0.087 | 0.162 | 0.171 | 0.044 | 0.012 | 0.453 | 0.068 | 0.056 | 0.069 | -5.806 | 2.412 |
| | 2008 | 0.075 | 1.116 | -0.005 | -0.272 | -0.728 | -0.185 | 0.196 | 0.192 | 0.06 | 0.021 | 0.412 | 0.09 | 0.083 | 0.1 | -3.204 | 4.307 |
| 5 | 2009 | -0.022 | 1.283 | -0.126 | -0.317 | -0.797 | 0.086 | 0.153 | 0.087 | 0.007 | 0.043 | 0.444 | -0.006 | 0.02 | 0.011 | -2.959 | 3.357 |
| | 2010 | -0.013 | 1.195 | -0.041 | -0.311 | -0.761 | 0.012 | 0.142 | 0.052 | 0.028 | -0.03 | 0.435 | 0.004 | 0.027 | -0.003 | -2.341 | 2.985 |
| | 2011 | -0.078 | 1.364 | -0.196 | -0.52 | -0.876 | 0.183 | 0.189 | 0.26 | 0.002 | 0.147 | 0.499 | -0.049 | -0.003 | -0.068 | -4.149 | 6 |
| | 2012 | 0.194 | 1.351 | 2.444 | -0.452 | -0.833 | 0.249 | 0.174 | 0.279 | 0.445 | 0.125 | 0.525 | 0.132 | 0.069 | 0.111 | -7.836 | 0.746 |
| | 2013 | 0.086 | 1.108 | -0.144 | -0.345 | -0.881 | 0.196 | -0.068 | 0.009 | 0.182 | 0.135 | 0.313 | 0.048 | -1.254 | 0.044 | -1.202 | -1.686 |
| 6 | 2014 | 0.092 | 1.176 | -0.012 | -0.237 | -0.85 | 0.21 | 0.08 | -0.128 | 0.151 | 0.156 | 0.301 | 0.059 | -0.968 | 0.055 | 2.152 | -1.717 |
| | 2015 | 0.1 | 0.889 | -0.108 | -0.179 | -0.981 | 0.188 | 0.111 | 0.009 | 0.059 | 0.139 | 0.268 | 0.072 | 0.868 | 0.057 | 3.483 | -0.209 |
| | 2016 | 0.087 | 0.846 | -0.11 | -0.185 | -0.929 | 0.189 | 0.359 | -0.04 | 0.26 | 0.109 | 0.307 | 0.053 | 1.674 | 0.042 | 3.757 | -8.132 |
| | 2017 | -0.519 | 0.764 | -0.889 | -0.224 | -1.07 | 0.217 | 0.243 | -0.018 | 0.326 | 0.298 | 0.511 | 0.26 | 1.493 | -0.829 | 3.517 | -25.771 |
| | 2018 | -6.982 | 0.464 | -0.604 | -0.203 | -0.742 | -0.253 | 0.269 | 0.042 | 0.156 | 0.231 | 1.864 | 0.104 | 0.562 | -0.312 | 1.532 | -14.85 |

**Fig 6. Decoupling of variables.**

policies resolve the CO2 emissions with a dynamic revolution of techniques [56, 57]. China is resolving the CO2 emissions issues from urbanization, population, and industrialization. And reduce the energy consumption demand from industrialization [29]. In 2020, CO2 emissions volume decrease by 40–45%, and imagine reducing by 60–65% by 2030 [58].

## 3.2 Energy consumption

Energy consumption entailed PEC, OIC, NGC, COC, NUC, HCC, and RGGB. The covariance of PEC indicated that a 1% change in PEC should get the variation of 0.00023% change in CO2 emissions because the covariance of beta is 1.0524%. The proportion of covariance calculated by PEC and its percentage change in five years. The PEC change is 0.0765% as for the beta proportion. It mostly changes by 1.0524% in CO2. The 1% change in CO2 emissions records at 0.0765%, as the same reduction has taped at 0.061% with a positive attitude.

The decoupling access by a 1% change in CO2 emissions minus from the computed beta value. We expected that the beta value changes 1.0524% by a positive attitude. However, the reality has happened that 0.0765% change in CO2 emissions. Therefore, the decoupling was -1.003, and in the next year -1.123. The additive result of both decoupling value -2.126, it shows that reduction in PEC. Besides that, CO2 emissions are increasing, and we compute decoupling it for the next five years. The decoupling value efficiently is recorded above -2. And it represents that CO2 emissions increased cause of PEC from 1989 to 1993 [52]. They estimated that the primary energy consumption (PEC) major driving force stimulates CO2 emissions in China. Industrial structure enhancing the energy system by import and export trade still increasing clean energy can contain CO2 emissions.

Fig 6 shows the periods stand in five years and we compute decoupling for each period from 1989 to 2018. Arrow indications show a change in valuation, increase and decrease by yellow, light green, and red. However, over 1% change in decoupling is showing by light green, and less than -1% change is showing by purple. **Note:** Variable's definition stated in Table 1.

According to Statista, primary energy consumption (PEC) of China increased by 4.18% from 2018, and in the last ten years, it recorded at 31.17%. In 2019, it amounted to 141.7 EJ (million exajoules) and 66.79% greater than from the USA. Additionally, China is one of the largest energy consumers such as 14 (million barrels daily) Oil energy (OIC), 307.3 (billion cubic meters) natural gas (NGC), 82 EJ coal (COC), 3.1 EJ nuclear power (NUC), 11.32 EJ

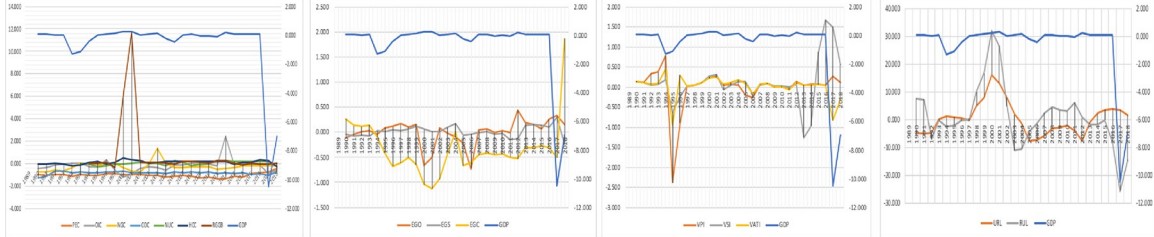

**Fig 7. GDP with primary energy consumptions, energy generation, value-added primary industries, and urban and rural population.**

hydroelectricity (HCC), 6.63 EJ renewable energy (RGGB) consumption recorded in 2018–2019 [59]. The study shows that the volume of PEC, OIC, NGC, COC, NUC, and HCC consumption raised in 6 periods and still rising in coming years.

The expected decoupling of the PEC, OIC, NGC, COC, NUC, HCC and RGGB recorded -0.464, -0.604, -0.203, -0.742, -0.253, -0.269 and 0.042 in the period of 2017 to 2021. Furthermore, the estimated results show the record volume of $CO_2$ emissions by % change during 2017 and 2018 with -1.228, -1.493, -0.427, -1.812, -0.036, -0.026 and 0.024. However, the $CO_2$ emissions will increase with the consumption of PEC, OIC, NGC, COC, NUC, and HCC, except for RGGB.

The renewable energy resources of biomass decoupling show a positive attitude every five periods from 1989 to 2018. However, the $CO_2$ emission will reduce by renewable biomass. The primary energy consumption (PEC, OIC, NGC, COC, NUC, HCC, and RGGB) with GDP has been plotted graphically in Fig 7.

The high-low line from 1999 to 2002 shows the highest positive attitude of RGGB with GDP, where the $CO_2$ emission had reduced by the renewable biomass as for the alternative causes of energy. As well as NGS and OIC showed positive decoupling from 2003 to 2005 and 2011 to 2012. $CO_2$ emissions reduce by oil and nuclear energy consumption.

The primary energy consumption quantifies and evaluates the energy or material metabolism. And environmental loads of oil consumption through the establishment and use of petroleum flow and energy flow on the $CO_2$ emissions [60]. Currently, China has evacuated the second-largest petroleum processor and cause $CO_2$ emissions, nearly a 60% increase in petroleum refining during 1990–2015 [61].

In 2009, China's automobile industries and vehicle sale excess from USA vehicle sales. And it befits number one automobile in the world. It's relatively connected with the demand for consumption oil per vehicle fuel consumption rates, which will effectively increase oil demand and consumption and $CO_2$ emissions by Chinese automobiles sectors [62, 63]. The combustion of biomass and fossil fuels contributes about 44% (range: 36 to 78%) of the total global $CO_2$ emissions [64]. The consumption of natural gas has a significant negative impact on $CO_2$ emissions, which indicates that a 1% change (increase) in natural gas will decrease $CO_2$ emissions by 0.0549% [65]. The advanced technologies make better policies and strategies for the utilization of energy. And also the use of appropriate and clean energy in China [66].

China's dynamic strategies are improving consistently. The generation of coal structure for the reduction of $CO_2$ emissions. Also, modified new techniques from using less-polluting and renewable (wind, water, solar, and nuclear) energies. However, nuclear energy sources are imperfect by public established impressions and modernization to utilize by the safe side. Biomass energy is also clean energy that can consistently use to achieve energy problems for the sustainability of an environment [36, 67].

### 3.3 Generation of electricity

It elicits electricity generation with EGO, EGS, and EGC. It shows the EGO covariance that a 1% change in EGO should get a change of 0.00065% change in $CO_2$ emissions because the covariance of beta is 0.0217%. The covariance of proportion calculates by EGO and its percentage change in five years. EGO change is 0.2301%, and as for the beta proportion, it most changes by 0.0217% in $CO_2$. 1% change in $CO_2$ recorded 0.2301%, so we have recorded some reduced value in decoupling is -0.2636% with a negative attitude. Decoupling access by a 1% change in $CO_2$ subtracted from the computed beta value. Therefore, we expected that the beta value changes 0.0217% by a positive attitude.

However, the reality happened that 0.2301% change in $CO_2$ emissions. The decoupling recorded 0.028, and in the next year with -0.046, the addition of both decoupling values is -0.018 in 1 to 2 period. It showed that $CO_2$ emissions increased because of electricity generation by oil (EGO), and as same we process as for each period. The estimated results of decoupling from 2014 to 2015 show that 0.059 with decreeing effects in EGO. The EGS and EGC results show increasing effects by 0.0139 and -0.268. The $CO_2$ emissions volume estimated by % change with 0.210, 0.295, and -0.569 in EGO, EGS, and EGC.

The estimated decoupling value is above -2, represents the increases in $CO_2$ emissions caused by excessive consumption of EGO from 1989 to 1993. The estimated oil production recorded 2.749 million barrels per day (mB/d) and consumption 87.88% in 1989, and within the next five years, the consumption increased 103.10%, and volume of production was only 5.30% [68]. Fig 7 shows electricity production by oil, gas, and coal (EGO, EGS, and EGC) with GDP. And the electricity production by EGO shows high-low line intensity with a positive attitude the $CO_2$ emission reduced from 2011 to 2013.

EGC shows a positive attitude of decoupling from 2017 to 2018, where the $CO_2$ emission had reduced by EGC. Many comprehensive studies have analyzed the relationship between energy consumption with economic growth in multiple countries. Their possible results divinely revealed that energy realistically is a significant part of economic growth [69, 70]. In China, electricity generation is contingent on coal, which has led to an increase in $CO_2$ emissions. [71] estimated that coal reached (+0.7%), crude oil (+6.8%), and natural gas (+8.6%) with uncertainty global $CO_2$ projection 0.8, 6.9, and 9.1 percent, while natural gas shows the highest projection volume. The coal demand increases by 1.0% (tonnes by weights) the remaining volume adjusts by stock change and import. The estimated results of electricity production presented that China using an alternative way to produce electricity for the high volume of $CO_2$ emission, energy consumption results estimated the coal was 80.2 EJ, oil 25.7 EJ, and natural gas 10.6 with a share of 59.0%,18.9%, and 7.8% during 2018. China's half coal used for electricity generation and demand for coal increased from coal-intensive industries of steel, cement, and non-ferrous metals in 2019.

Because of high pressure, China cut its $CO_2$ emissions from all sources of coal and announced it would cut $CO_2$ emissions (per-unit) of GDP by 40% to 45% in 2020 from 2005 [40, 72, 73]. While as for the reduction of $CO_2$ emissions, environmental quality and water shift from coal to shale gas will improve consumption and public health [74, 75].

### 3.4 Value-added services

The value-added services of the industries obtain by primary, secondary, and tertiary manufacturing industries with VPI, VSI, and VATI. It shows the covariance of VPI that a 1% change in VPI should get a change -0.000298% change in $CO_2$ emissions because the covariance of beta is -0.3245%. The proportion of covariance calculated by VPI and its percentage change in five years. VPI change is 0%, as for the beta proportion, it most changes by -0.3245%

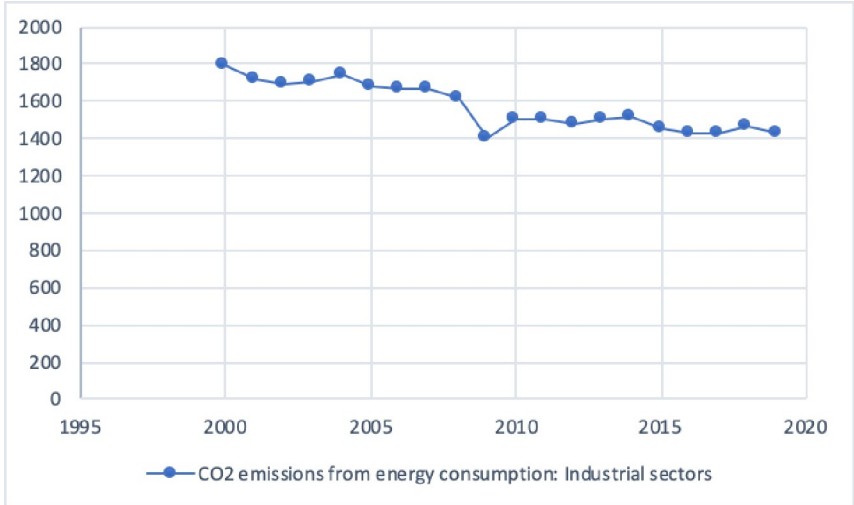

**Fig 8. CO2 emission industrial sectors (a million tons).**

in CO2. 1% change in CO2 recorded 0%, so we have recorded some increased in decoupling value is 0.0394% with a positive attitude. Decoupling access by a 1% change in CO2 subtracted from the computed beta value.

Therefore, we expected that beta value changes -0.3245% by a negative attitude. Estimated change in CO2 emissions by 0% and the decoupling recorded at 0.374, and in the next year with 0.785 with increasing effects. We record the additive value of both decoupling 1.159 in 1 to 2 periods. Same as all reaming periods. The highest estimated decoupling results are recorded from 2017 to 2018 show that 0.104 and 0.562 in VPI and VSI with decreasing effects. And the VATI results show increasing effects by -0.312, and the CO2 emissions volume increased by VPI and VSI.

The expected decoupling value of the VPI, VSI, and VATI occurred as 0.104, 0.562, and -0.312 from 2017 to 2021. However, the CO2 emissions will increase from the value-added territory industries (VATI) and reduce by VPI and VSI. Value-added industries results show a positive attitude in one to six periods. However, CO2 emissions reduce by VPI. CO2 emissions have increased in the 5th and 6th periods of VSI and VATI.

Fig 8 shows that at the end of 2019, 1,423 (a million tons) CO2 emission have recorded by energy consumption in industrial sectors. It is significant that cut CO2 emissions from energy consumption by 2030 goals [76].

The value-added industries of the primary, secondary, and territory (EGO, EGS, and EGC) with GDP plotted in Fig 7. As a prior study, 80% of energy consumption has contributed to the industrial sectors of China's total emission since 2005 [77]. In 2014, China overtook the USA as the world's biggest CO2 emitter [78]. The estimated value recorded that China (9,135 million tons), 177% of the USA emissions, and 28.21% of global emissions [79]. Energy consumption and CO2 emissions accounted for about 68% and 84% in China in 2015 [80]. Chinese industrial sectors have been the focus on several policies to control and improve energy efficiency. The China Industrial Green Development Plan (CIGDP) targeted to reduce the volume of CO2 emission and intensity by 22% and 18%, and respectively 34% and 40% reduce until 2025 [81]. Many studies examined the value-added industries of China. This study has analyzed beta decoupling techniques for six periods. A high and low line from 2014 to 2018 shows the highest positive attitude of VSI with GDP, where the CO2 emission had reduced by the value-added industries from the territory (VATI). The VPI showed a positive decoupling

between 1991 and 1994, and the volume of $CO_2$ emission has reduced by energy generation from primary value-added industries.

## 3.5 Urbanization and ruralization

It elicits the urbanization and ruralization with URL and RUL. It shows the covariance of URL that a 1% change in URL should get change -1.4105e-05 change in $CO_2$ emissions because the covariance of beta is -0.0967%. The proportion of covariance calculates by URL and its percentage change in five years.

The URL change is 0.0309%, and the beta proportion mostly changes by -0.0967% in the level of $CO_2$ emissions. The 1% change in $CO_2$ record 0.0309%, so we have recorded some reduced value in decoupling is 0.0300 with a positive attitude. And the decoupling access by a 1% change in $CO_2$ emissions minus from the computed beta value. Therefore, we expected that beta value changes -0.0967%.

By the negative attitude, the reality happened that 0.0309% change in $CO_2$ emissions, and the decoupling occurred 0.146 in 1 period. In the next year, increase with 0.979. Additive results show both decoupling is 1.125. And the estimated value is showing that $CO_2$ emissions increase because of the URL. Same as that, we compute every five years $CO_2$ emissions [82]. Represented that China's urbanization level increased 20.61% from 2009 to 2019. Every year it increases over 1% and 60.31% people lived in cities and urban areas from the total population. China aim to adjust 70% urbanization by 2030. However, the urbanization is not only current miracles rejuvenation but also industrialization sped up. It progresses potentially over 100 million people transited to Chinese different cities by 2020. However, it is likely leading to large increase in $CO_2$ emissions and energy consumption [83, 84]. This study recorded the expected decoupling result 1.532 in URL with decreasing effects and -14.850 in RUL. The $CO_2$ emissions volume estimated by % change of 5.049 and -40.621 in URL and RUL from 2017 to 2021.

However, the $CO_2$ emissions will increase from the ruralization (RUL) and reduce by urbanization (URL). Urbanization decoupling shows a positive attitude in 1st, 2nd, 3rd and 6th periods. In this case, the $CO_2$ emissions are reduced by the effect of URL. The $CO_2$ emissions have increased in 4th and 5th periods in URL and 1st, 3rd, 5th and 6th periods of RUL Fig 6. The urbanization and ruralization (URL and RUL) with GDP plotted in Fig 7.

The High-low line from 1997 to 2002 shows the highest positive attitude of RUL with GDP. In this stage, the value of $CO_2$ emissions reduced by the ruralization of the population. The URL showed positive decoupling between 1997 and 2004. The $CO_2$ emissions intensity reduces by urbanization. As prior research, China contributed approx. 10% of global greenhouse gas emissions in 2017, where 71% of global $CO_2$ emission contributed from the urban areas, because of high-tech and socioeconomic activities [30].

The environment influences China's urbanization 14,000 $Km^2$ areas experienced a high concentration of $CO_2$ emissions [85]. The rapid dimension in construction and renovation of space heating is obsessed with urbanization [86, 87]. The higher intensity of $CO_2$ emissions is recorded in the urban territories than residential or rural areas because of the higher volume of traffic and denser population [88, 89].

Fig 9 represents a 1% change in each period with the effect of a change in valuation, increase, and decrease by yellow light green, and red arrow. The light blue, yellow colors show the top and bottom changes by 10% in different periods. A 1% change in NUC (0.7248), HCC (0.3126), and RGGB (0.5072) is revealing a 1% change in $CO_2$ emissions by positive decoupling in the third and fourth periods. Besides, the pivot chart is showing a sum of decoupling with GDP effects. It is representing a 1% change in $CO_2$ emissions influence by decoupling from 1989 to 2018. We record the highest decoupling intensity on RUL. It represents the

| Years | CO2 | GDP | PEC | OIC | NGC | COC | NUC | HCC | RGGB | EGO | EGS | EGC | VPI | VSI | VATI | URL | RUL |
|---|---|---|---|---|---|---|---|---|---|---|---|---|---|---|---|---|---|
| | I | A | T1 | T2 | T3 | T4 | T5 | T6 | T7 | T8 | T9 | T10 | IS1 | IS2 | IS3 | UL1 | UL2 |
| 1989-1993 | 0.0808 | 0.1388 | 0.0765 | 0.1052 | 0.0617 | 0.0652 | | 0.1619 | | 0.2301 | 0.2663 | 0.0386 | 0.0000 | 0.0000 | 0.0000 | 0.0310 | 0.0041 |
| 1994-1998 | -0.0009 | 0.0785 | 0.0013 | 0.0273 | 0.0312 | -0.0076 | 0.0221 | 0.0148 | 0.0889 | 0.1332 | 0.2458 | 0.0284 | 0.0343 | 0.0890 | 0.0840 | 0.0547 | 0.0122 |
| 1999-2003 | 0.1792 | 0.1004 | 0.1650 | 0.1134 | 0.1619 | 0.1918 | 0.7248 | 0.0149 | 0.0039 | 0.0813 | 0.2291 | 0.1851 | 0.0237 | 0.1267 | 0.0954 | 0.0431 | 0.0178 |
| 2004-2008 | 0.0191 | 0.0965 | 0.0376 | 0.0205 | 0.1527 | 0.0158 | 0.1008 | 0.3126 | 0.5072 | 0.3305 | 0.0194 | 0.0197 | 0.0516 | 0.2114 | 0.2107 | 0.0292 | 0.0153 |
| 2009-2013 | 0.0274 | 0.0776 | 0.0386 | 0.0444 | 0.1392 | 0.0214 | 0.1460 | 0.0543 | 0.2231 | 0.0845 | 0.0557 | 0.0848 | 0.0381 | 0.0799 | 0.0830 | 0.0271 | 0.0196 |
| 2014-2018 | 0.0216 | 0.0694 | 0.0428 | 0.0500 | 0.1770 | 0.0086 | 0.1865 | 0.0321 | 0.1398 | 0.0000 | 0.1026 | 0.0645 | 0.0174 | 0.0506 | 0.0983 | 0.0220 | 0.0219 |

**Fig 9. Percentage change of indicators.** Note: Variable's definition stated in Table 1.

percentage change in CO2. However, it might have affected by a tremendous level of urbanization to ruralization Fig 10.

The α and β results are impeaching foresee, and compare the estimated value of a 1% change in indicators, change in CO2 emissions because the covariance of β shows a positive or negative attitude. We calculate the proportion of covariance by each variable and its percentage of change in five years in each period. Fig 11 is comparing the five-year estimated results of all variables with the predicted value. A measure portfolio aptitude with a 1% change in GDP prefers to invest in each variable by higher α. The 1% change in CO2 emissions results from the α propensity, and the investor use β to determine how much downside expects by a difference in investment in indicators.

Evaluation of β analyzed by risk-averse on low and take the target on higher β hope to come for volatility. The URL (3.2863), PEC (1.1015), and VSI (1.2481) show higher results, and their decoupling indicated that we should reduce CO2 emissions by PEC, VSI, and URL. Because a 1% change in indicators has to change in emission level and created volatility. Fig 12 shows the predicted results of each period with top-bottom 10% results. The 1% change in PEC, VSI, and URL regarding -0.005, 0.1289, and 0.2754, should receive a change in CO2 emissions.

## 4. Conclusions

In this study, we used the Beta Decoupling Techniques (BDT) to calculate the CO2 emissions by GDP, energy consumption (PEC, OIC, NGC, COC, NUC, HCC, and RGGB), electricity generation (EGO, EGS, and EGC), value-added industries (VPI, VSI, and VATI), and population (URL and RUL) of China from 1989 to 2018. We collect the relevant energy data set from

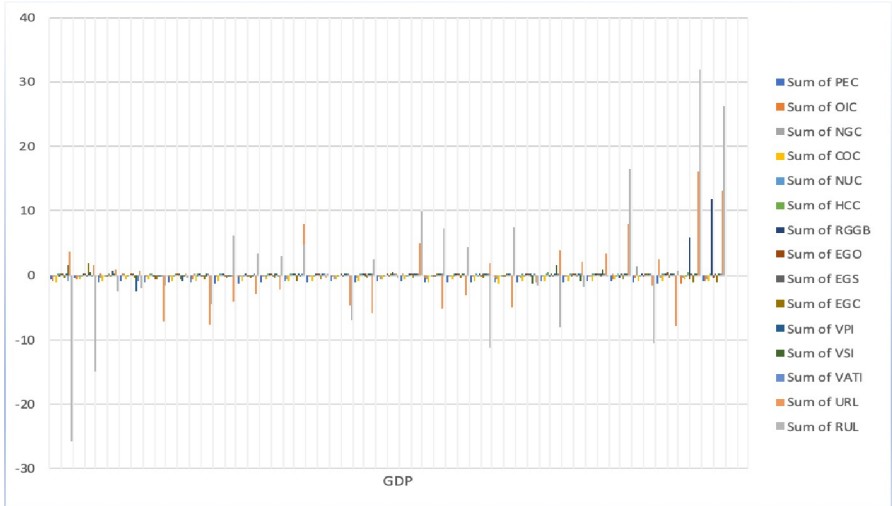

**Fig 10. Pivot chart with GDP.**

| Years | GDP | PEC | OIC | NGC | COC | NUC | HCC | RGGB | EGO | EGS | EGC | VPI | VSI | VATI | URL | RUL |
|---|---|---|---|---|---|---|---|---|---|---|---|---|---|---|---|---|
| | A | T1 | T2 | T3 | T4 | T5 | T6 | T7 | T8 | T9 | T10 | IS1 | IS2 | IS3 | UL1 | UL2 |
| 1989-1993 | 0.0039 | 1.0524 | 0.0963 | 0.6995 | 0.6952 | - | 0.0689 | - | 0.0217 | 0.1243 | 0.0934 | 0.3246 | 0.0198 | 0.0421 | 0.0967 | 0.4591 |
| 1994-1998 | -0.0008 | 0.9827 | -0.2871 | 0.2748 | 0.8096 | 0.2287 | 0.1129 | 0.0332 | 0.0605 | 0.0154 | 0.5574 | -0.001 | 0.0023 | 0.0003 | 4.8901 | -9.89 |
| 1999-2003 | 0.0161 | 1.1406 | 0.3947 | 0.1409 | 0.9133 | 0.0455 | 0.1053 | 0.0086 | 0.1171 | 0.0033 | 0.5046 | 0.0368 | 0.0526 | 0.0221 | 1.7629 | 1.3248 |
| 2004-2008 | 0.0075 | 1.1986 | 0.087 | 0.3546 | 0.8104 | 0.2677 | 0.1137 | 0.1094 | 0.022 | 0.0619 | 0.494 | 0.0077 | 0.0004 | 0.0175 | 3.2863 | 4.2242 |
| 2009-2013 | -0.0917 | 1.1015 | 0.1383 | 0.3385 | 0.8752 | 0.2021 | 0.0616 | 0.0151 | 0.1877 | 0.1414 | 0.3071 | 0.0541 | 1.2481 | 0.0505 | 1.1963 | 1.68 |
| 2014-2018 | 6.9815 | 0.4641 | 0.6039 | 0.2026 | 0.7421 | 0.2528 | 0.2691 | 0.0416 | 0.1562 | 0.2309 | 1.8636 | 0.1036 | 0.5623 | 0.3122 | -1.532 | 14.85 |

**Fig 11. Five years beta.** Note: Variable's definition stated in Table 1.

the National Bureau of Statistics of China from energy consumption, industrialization, and population.

We analyzed the decoupling relationship between carbon emission, gross domestic product, primary energy consumption, electricity generation, value-added industries, and population of each period in China with the BDT. The top 10% results analyze the RGGB, EGO, EGS, VPI, VATI, URL, and RUL. In the 3rd and 6th periods, the EGC and RUL estimated value is over 10% on the primary energy consumption, electricity generation, value-added industries, and population from 1989 to 2018. It shows a trend of increase with time. The EGC and RUL still accounted for a large proportion in 2017–2021 and over 90% of the total energy consumption.

The above result shows the energy consumption resources in China. From the eventual overall decoupling effect on energy consumption, electricity generation, and urbanization. We calculate the decoupling state based on the Beta Decoupling Techniques (BDT). This study analyzed the economic growth of energy consumption, industrial production, electricity generation, and population. Additionally, China achieves a robust decoupling state and less dependent on renewable and fossil vitality. Also, China has the countless ability to assurance resource security. The decoupling significantly overestimated the energy level in China, especially on the industrial and ruralization side. The decoupling estimation of unique sources of energy concluded that the dependence of China's rapid economy on a different level of energy sources differed from the perspective of energy consumption, electricity generation, and urbanization.

According to the above results, the decoupling in 2nd (NUC, RGGB, and EGS), 4th and 5th (EGO), 3rd, 5th, and 6th (RUL) periods state that less than 10% results of negative decoupling. The CO2 emissions will be increases in each period, so we should reduce the energy consumption in the form of electricity generation from oil use and ruralization. The illustrated results show that CO2 emissions are less dependent on COC, NUC, RGGB, EGO, EGS, VSI, and VSTI in a different period. Based on the above decisions, this paper situates forward relevant strategies recommendations.

First, total energy consumption is vital issues that affect resource security and energy development strategies [90, 91]. It is most necessary to re-change growth and control by the level of energy. Besides, the population, solicitation of clean energy (solar, water, nuclear, wind, geothermal, tidal, and biomass), and to build the low carbon energy sources by technology. Also, develop renewable vitality mechanism aiming to improve the efficiency of utilization, energy conversation, energy transformation, and consumption [86, 92].

| Years | GDP | PEC | OIC | NGC | COC | NUC | HCC | RGGB | EGO | EGS | EGC | VPI | VSI | VATI | URL | RUL |
|---|---|---|---|---|---|---|---|---|---|---|---|---|---|---|---|---|
| | A | T1 | T2 | T3 | T4 | T5 | T6 | T7 | T8 | T9 | T10 | IS1 | IS2 | IS3 | UL1 | UL2 |
| 1989-1993 | 0.0532 | -0.0047 | 0.0455 | 0.0269 | - | - | 0.0469 | - | 0.0527 | 0.0483 | 0.0629 | 0.0681 | 0.0572 | 0.0587 | 0.0564 | 0.0524 |
| 1994-1998 | 0.0214 | -0.005 | 0.037 | -0.045 | 0.0181 | 0.0116 | 0.0115 | 0.0259 | 0.0209 | 0.0256 | -0.0152 | 0.0214 | 0.0217 | 0.0212 | 0.2754 | -0.1107 |
| 1999-2003 | 0.1346 | -0.0132 | 0.1012 | -0.0093 | 0.1304 | 0.1241 | 0.1266 | 0.1394 | 0.1406 | 0.1384 | 0.0596 | 0.1336 | 0.1289 | 0.1403 | 0.2063 | 0.3254 |
| 2004-2008 | 0.0559 | -0.0192 | 0.0522 | -0.0939 | 0.0423 | 0.0314 | 0.0689 | 0.0922 | 0.0611 | 0.0342 | 0.0076 | 0.0582 | 0.0581 | 0.0604 | -0.0564 | -0.0333 |
| 2009-2013 | 0.0166 | -0.0207 | 0.001 | -0.0735 | 0.0068 | 0.0475 | 0.0002 | 0.0098 | 0.0017 | 0.0239 | -0.0005 | 0.0122 | -0.0823 | 0.0141 | -0.0264 | 0.0428 |
| 2014-2018 | -0.4447 | -0.0001 | -0.0104 | -0.1037 | 0.0157 | -0.0274 | 0.0112 | 0.0256 | 0.0198 | -0.0039 | 0.1401 | 0.018 | 0.0483 | -0.0109 | 0.0535 | 0.3443 |

**Fig 12. Five years alpha.** Note: Variable's definition stated in Table 1.

Second, each country needs to take serious action against CO2 emissions [93]. Therefore, they can take measures to energy demand whenever planning to develop enormous industries and control emissions reduction measures with adapting technology.

## 5. Policy recommendation

As for the development strategies, the government should expand its investment in the capital-intensive primary, secondary, and territory industries and promote the green technology emerging industries by enhancing technological innovations. The focus of reducing the growth rate of energy consumption should be on improving energy efficiency use. Also, control and change strategies in natural-gas resources and how to secure and ensure the safe supply of natural gas in China should modernize and focus on the tremendous macro policy.

Contemplating that the industrial structure, related it to GDP, the prominence of coming research ought to be on sightseeing the relationship between energy and economic growth from the outlook of primary energy consumption, electricity generation, industries, and population.

We recommend it to take optimization and amendment policies for demand and supply structure to provide cherished strategies. Besides, adjusting and enhancing the economic sustainability and development structure of China.

## Supporting information

**S1 File.**
(DOCX)

**S2 File.**
(DOCX)

**S3 File.**
(XLS)

## Acknowledgments

I address the valuable direction, and I cannot express enough gratitude to my teachers for their support and reassurance. I extend my heartfelt affection for the learning opportunities granted by the Fujian University of Technology. I appreciate my parents and family who encourage me to as a Ph.D. scholar. I cannot forget the importance of the impaired scholar who scarifies all emotions and sentiments for their bright future. I recognize to be-love supporter who holds me and gives me strength and power for extant and enduring.

## Author Contributions

**Conceptualization:** Rabnawaz Khan.

**Data curation:** Rabnawaz Khan.

**Formal analysis:** Rabnawaz Khan.

**Funding acquisition:** Rabnawaz Khan.

**Investigation:** Rabnawaz Khan.

**Methodology:** Rabnawaz Khan.

**Project administration:** Rabnawaz Khan.

**Resources:** Rabnawaz Khan.

**Software:** Rabnawaz Khan.

**Supervision:** Rabnawaz Khan.

**Validation:** Rabnawaz Khan.

**Visualization:** Rabnawaz Khan.

**Writing – original draft:** Rabnawaz Khan.

**Writing – review & editing:** Rabnawaz Khan.

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
