## [Decision Letter · Decision Letter 0]

21 Dec 2020

PONE-D-20-34342

Beta decoupling relationship between CO2 emissions by GDP, energy consumption, electricity production, value-added industries, and population in China

PLOS ONE

Dear Dr. Khan,

Thank you for submitting your manuscript to PLOS ONE. After careful consideration, we feel that it has merit but does not fully meet PLOS ONE’s publication criteria as it currently stands. Therefore, we invite you to submit a revised version of the manuscript that addresses the points raised during the review process.

We look forward to receiving your revised manuscript.

Kind regards,

Bing Xue, Ph.D.

Academic Editor

PLOS ONE

Journal Requirements:

"I acknowledge the helpful instruction and comments from my supervisor and thanks to the

financial support of the National Natural Science Foundation of China of the Fujian University

of Technology."

"Yes."

"No."

4. Please include a caption for figure 8.

Reviewers' comments:

Reviewer's Responses to Questions

**Comments to the Author**

1. Is the manuscript technically sound, and do the data support the conclusions?

Reviewer #1: Partly

2. Has the statistical analysis been performed appropriately and rigorously? 

Reviewer #1: Yes

3. Have the authors made all data underlying the findings in their manuscript fully available?

Reviewer #1: Yes

4. Is the manuscript presented in an intelligible fashion and written in standard English?

Reviewer #1: Yes

5. Review Comments to the Author

Reviewer #1: 1 The abstract can be further refined.

2 There are more than five keywords in the abstract.

3 The introduction does not elaborate on the topic of this article and its value, and we recommend rewriting.

4 The contribution of this article is not very obvious, and it is recommended to rewrite.

5 In the part 3, the reasons for the research results need to be explained and demonstrated.

6. PLOS authors have the option to publish the peer review history of their article (what does this mean?). If published, this will include your full peer review and any attached files.

Reviewer #1: No

---

## [Author Response · Author response to Decision Letter 0]

22 Jan 2021

Authors' Response to Reviewers' Comments

No: [PONE-D-20-34342]

Title: Beta decoupling relationship between CO2 emissions by GDP, energy consumption, electricity production, value-added industries, and population in China 

Dear editor and reviewers,

Thank you very much for your letter and reviewers’ valuable suggestions on our manuscript entitled “Beta decoupling relationship between CO2 emissions by GDP, energy consumption, electricity production, value-added industries, and population in China” No: [PONE-D-20-34342]. Those comments are very valuable and helpful for improving our paper, as well as providing the important guiding significance to our research. According to the editor’s and reviewers’ comments, we have carefully checked and improved the manuscript. The questions in the comments were marked in blue, and the corresponding responses are listed below point by point. 

Response to additional requirements. 

https://journals.plos.org/plosone/s/file?id=wjVg/PLOSOne_formatting_sample_main_body.pdf andhttps://journals.plos.org/plosone/s/file?id=ba62/PLOSOne_formatting_sample_title_authors_affiliations.pdf

Reply: Thank you very much for your valuable comments and suggestions. According to your valued comments, we have revised it carefully as follows. 

We have revised the paper point by point by the valuable comment.

According to the Plos One manuscript body formatting guideline, I modified the whole paper, including heading, figures, and table style. The complex structure of Beta decoupling techniques (BDT) tables 2, 3,4, and 5 do not fulfill the requirement of table style, therefore as per given instruction all the tables are converted into figure caption like 6,9,11, and 12.

2. Thank you for stating the following in the Acknowledgments Section of your manuscript: "I acknowledge the helpful instruction and comments from my supervisor and thanks to the financial support of the National Natural Science Foundation of China of the Fujian University of Technology." We note that you have provided funding information that is not currently declared in your Funding Statement. However, funding information should not appear in the Acknowledgments section or other areas of your manuscript. We will only publish funding information present in the Funding Statement section of the online submission form. Please remove any funding-related text from the manuscript and let us know how you would like to update your Funding Statement. Currently, your Funding Statement reads as follows:

"Yes."

Reply: Thank you very much for your valuable comments and suggestions. According to your valued comments, we have revised it carefully as follows. 

We have revised the paper point by point by the valuable comment.

As per the given instruction, I have excluded the funding information from acknowledgment (Line 28). 

4. Please include a caption for figure 8.

Reply: Thank you very much for your valuable comments and suggestions. According to your valued comments, we have revised it carefully as follows. 

We have revised the paper point by point by the valuable comment.

According to the Plos One manuscript body formatting guideline, I modified the figures and table captions under the stated requirements. Tables exported as TIFF figure, all the figure captions have been revised and cite and re-label. Figure 8 caption is updated now it shows figure 10. 

Reviewers' comments: 

5.Reviewer's Responses to Questions

Reviewer #1: 1 The abstract can be further refined.

Reply: Thank you very much for your valuable comments and suggestions. According to your valued comments, we have revised it carefully as follows. 

We have revised the paper point by point by the valuable comment.

As per the given instruction, I have modified and refined the abstract and make it easy for the reader. likewise 

"The sources of fossil energy are a vital cause of economic growth and influence on security, whereas energy enhances or promotes socio-economic stability and the environment. As the fossil energy sources supply has become progressively stern, reconnoitering the beta decoupling relationships between CO2 emissions, GDP, energy consumption, electricity consumption, value-added industries, and population in China, and the result will be favorable for illustrative the security of the resources. This study adopts the extended stochastic model (STIRPAT) with Beta Decoupling Techniques (BDT). This technique employs the decoupling situation by the alpha and beta effects from 1989 to 2018 and calculates the % change in CO2 emissions by GDP growth and energy consumption. The estimated results are negative and economic growth depends on coal and natural gas. First CO2 emissions annually increasing cause of rapid growth, energy consumption, and electricity production, and the structural contradiction of energy remained static. Second, the Value-added industries estimated that CO2 emissions reduce by primary industries. Third, the decoupling states of CO2 emissions and population show an inverse relationship. This paper suggests that China is sustainable and strengthen energy output, transmute the energy consumption structure, and advance development policies under the environmental circumstance."

2 There are more than five keywords in the abstract.

Reply: Thank you very much for your valuable comments and suggestions. According to your valued comments, we have revised it carefully as follows. 

We have revised the paper point by point by the valuable comment.

As per the given instruction, I have used only the five most relevant and authentic words as keywords. Likewise, the Beta Decoupling Technique, stochastic model (STIRPAT), energy consumption, electricity production, and value-added industries.

3 The introduction does not elaborate on the topic of this article and its value, and we recommend

rewriting.

Reply: Thank you very much for your valuable comments and suggestions. According to your valued comments, we have revised it carefully as follows. 

We have revised the paper point by point by the valuable comment.

As per the given instruction, the first section introduction is modified and improved with new research findings. Modified article lines are stated below. 

Modified lines with track change: 

1. Introduction: (237 to 253), (257 to 263), (266 to 270), (273 to 278), (319 to 353)

1.2 Primary energy consumption: (360 to 364) 

1.3 Electricity generation: (386 to 391)

1.4 Value-added industries: (240 to 248)

4 The contribution of this article is not very obvious, and it is recommended to rewrite.

Reply: Thank you very much for your valuable comments and suggestions. According to your valued comments, we have revised it carefully as follows. 

We have revised the paper point by point by the valuable comment.

As per the given instruction, the contribution of the article rewrites from the line 319 to 353. 

A significant contribution of this study to the existing research is the Beta decoupling technique (BDT). (1) Mostly, existing studies sought causes for decoupling economic upheaval from CO2 emissions based on technological changes and production. (2) It used the decomposition approach because of the logarithmic mean division index (LMDI), which shows the index decomposition method. It limits the LMDI approach in case of technical efficiency and growth [24]. Therefore, it’s needed to introduce a more meaningful technique to determine the significant effects of CO2 emissions by growth and energy consumption. (3) The Beta decoupling technique expanded by planned (IPAT) into a stochastic model (STIRPAT) analysis of CO2 emissions. (4) We examined the decoupling among CO2 emissions, GDP, energy consumption, electricity production, value-added industries, and population for China from 1989 to 2018, while most prior studies have explored erstwhile to 1989. This research study should specify a thoughtful sound of Beta decoupling and its causes for the latest eras in China.

5 In the part 3, the reasons for the research results need to be explained and demonstrated.

Reply: Thank you very much for your valuable comments and suggestions. According to your valued comments, we have revised it carefully as follows. 

We have revised the paper point by point by the valuable comment.

As per the given instructions, the third part of the article amends and modified. The research results are explained and demonstrated line by line. 

Modified lines with track change: 

3.1 Gross development product (GDP): (740 to 750), and (757 to 774)

3.2 Energy Consumption: (793 to 796), (833 to 840), (844 to 845)

3.3 Generation of Electricity: (916 to 942)

3.4 Value-added Services: (1002 to 1008), (1024 to 1090)

3.5 Urbanization and Ruralization: (1110 to 1118)

Thank you very much again for your valuable comments and suggestions.

---

## [Decision Letter · Decision Letter 1]

18 Feb 2021

PONE-D-20-34342R1

Beta decoupling relationship between CO2 emissions by GDP, energy consumption, electricity production, value-added industries, and population in China

PLOS ONE

Dear Dr. Khan,

Thank you for submitting your manuscript to PLOS ONE. After careful consideration, we feel that it has merit but does not fully meet PLOS ONE’s publication criteria as it currently stands. Therefore, we invite you to submit a revised version of the manuscript that addresses the points raised during the review process.

We look forward to receiving your revised manuscript.

Kind regards,

Bing Xue, Ph.D.

Academic Editor

PLOS ONE

Reviewers' comments:

Reviewer's Responses to Questions

**Comments to the Author**

1. If the authors have adequately addressed your comments raised in a previous round of review and you feel that this manuscript is now acceptable for publication, you may indicate that here to bypass the “Comments to the Author” section, enter your conflict of interest statement in the “Confidential to Editor” section, and submit your "Accept" recommendation.

Reviewer #2: (No Response)

Reviewer #3: All comments have been addressed

2. Is the manuscript technically sound, and do the data support the conclusions?

Reviewer #2: Yes

Reviewer #3: Yes

3. Has the statistical analysis been performed appropriately and rigorously? 

Reviewer #2: Yes

Reviewer #3: Yes

4. Have the authors made all data underlying the findings in their manuscript fully available?

Reviewer #2: Yes

Reviewer #3: Yes

5. Is the manuscript presented in an intelligible fashion and written in standard English?

Reviewer #2: Yes

Reviewer #3: Yes

6. Review Comments to the Author

Reviewer #2: After a careful assessment of the manuscript, I believe this informative study is suitable for publication.

This article describes that the sources of fossil energy are a vital cause of economic growth and influence on security, whereas energy enhances or promotes socio-economic stability and the environment. As the fossil energy sources supply has become progressively stern, reconnoitering the beta decoupling relationships between CO2 emissions, GDP, energy consumption, electricity consumption, value-added industries, and population in China, and the result will be favorable for illustrative the security of the resources. This study adopts the extended stochastic model (STIRPAT) with Beta Decoupling Techniques (BDT). This technique employs the decoupling situation by the alpha and beta effects from 1989 to 2018 and calculates the percentage change in CO2 emissions by GDP growth and energy consumption.

Abstract and Introduction improvement:

I happy to evaluate this interesting study. In my opinion, I have some guidelines for the authors to enhance the study quality before endorsing it for publication. As the Abstract is the main door or "FACE" of the manuscript, it should briefly present high-quality English with new information. I have suggested some studies to check the abstracts, improve yours, cite them in the introduction, and build your study objectives like these studies.

Abbas, J., Mahmood, S., Ali, H., Ali Raza, M., Ali, G., Aman, J., Bano, S., & Nurunnabi, M. (2019). The Effects of Corporate Social Responsibility Practices and Environmental Factors through a Moderating Role of Social Media Marketing on Sustainable Performance of Business Firms. Sustainability, 11(12), 3434. https://www.mdpi.com/2071-1050/11/12/3434

Hussain, T., Abbas, J., Wei, Z., Ahmad, S., Xuehao, B., & Gaoli, Z. (2021). Impact of Urban Village Disamenity on Neighboring Residential Properties: Empirical Evidence from Nanjing through Hedonic Pricing Model Appraisal. Journal of Urban Planning and Development, 147(1), 04020055. https://doi.org/10.1061/(asce)up.1943-5444.0000645

Abbas, J., Raza, S., Nurunnabi, M., Minai, M. S., & Bano, S. (2019). The Impact of Entrepreneurial Business Networks on Firms’ Performance Through a Mediating Role of Dynamic Capabilities. Sustainability, 11(11). https://doi.org/10.3390/su11113006

Methods and Results

The results section of the paper presents a good view of the study. This work presents a notable investigation on a selected topic. I suggest including some graphical presentations to improve the quality of this study. Please see the proposed studies and see the graphical representation. Improve your work like these studies and cite them in this section.

Abbasi, K. R., Abbas, J., & Tufail, M. (2021, 2021/02/01/). Revisiting electricity consumption, price, and real GDP: A modified sectoral level analysis from Pakistan. Energy Policy, 149, 112087. https://doi.org/10.1016/j.enpol.2020.112087

Mubeen, R., Han, D., Abbas, J., & Hussain, I. (2020). The Effects of Market Competition, Capital Structure, and CEO Duality on Firm Performance: A Mediation Analysis by Incorporating the GMM Model Technique. Sustainability, 12(8). https://doi.org/10.3390/su12083480

Raza Abbasi, K., Hussain, K., Abbas, J., Fatai Adedoyin, F., Ahmed Shaikh, P., Yousaf, H., & Muhammad, F. (2021). Analyzing the role of industrial sector's electricity consumption, prices, and GDP: A modified empirical evidence from Pakistan. AIMS Energy, 9(1), 29-49. https://doi.org/10.3934/energy.2021003

Conclusion

I suggest you make a separate heading of the conclusion and do not mix it with implications.

Policy Recommendations

I again recommend you to make a separate heading of the Policy Recommendations.

The conclusion section is acceptable. Overall, this presents a good piece of research work. I recommend that authors do a little more work and revise this article accordingly. I suggest the authors check English quality and fix some weak sentences. If you have already taken English editing service, ask them to recheck the quality to meet scientific merit for publication. I endorse this manuscript for publication after minor corrections, as suggested.

Reviewer #3: Dear Authors,

Thank you for the opportunity for me to go through and read your Paper Titled: Beta decouplingrelationship between CO2 emissions byGDP, energy consumption, electricity production, value-added industries,and population in China.

The authors have done a great a job by revising the paper. The Abstract and Intruduction has been revised Accordingly. Also the Research Results is well explained and Demonstrated, therefore the recommend this paper for Publication.

7. PLOS authors have the option to publish the peer review history of their article (what does this mean?). If published, this will include your full peer review and any attached files.

Reviewer #2: No

Reviewer #3: No

---

## [Author Response · Author response to Decision Letter 1]

13 Mar 2021

Authors' Response to Reviewers' Comments

No: PONE-D-20-34342R1

Title: Beta decoupling relationship between CO2 emissions by GDP, energy consumption, electricity production, value-added industries, and population in China 

Dear editor and reviewers,

Thank you very much for your letter and reviewers' valuable suggestions on our manuscript entitled "Beta decoupling relationship between CO2 emissions by GDP, energy consumption, electricity production, value-added industries, and population in China”. No: [PONE-D-20-34342R1]. These comments are very esteemed and helpful for meaningfully improving our paper. It is providing the imperative guiding significance to our research. According to the editors' and reviewers’ comments, we have carefully checked and improved the manuscript. The questions in the comments were marked in blue, and the corresponding responses are listed below point by point.

Reviewers' comments: 

Response to additional requirements. 

6. Review Comments to the Author 

Reviewer #2: After a careful assessment of the manuscript, I believe this informative study is suitable for publication.

This article describes that the sources of fossil energy are a vital cause of economic growth and influence on security, whereas energy enhances or promotes socio-economic stability and the environment………. energy consumption.

Abstract and Introduction improvement: I happy to evaluate this interesting study. In my opinion, I have some guidelines for the authors to enhance the study quality before endorsing it for publication. As the Abstract is the main door or "FACE" of the manuscript, it should briefly present high-quality English with new information. I have suggested some studies to check the abstracts, improve yours, cite them in the introduction, and build your study objectives like these studies.

Reply: Thank you very much for your valuable comments and suggestions. According to your valued comments, we have revised it carefully as follows. 

We have revised the paper point by point by the valuable comment.

Under your kindful consideration, I have changed sentences with pure wording and increase reader interest. I have also read your research papers, and after profound review, I modified the abstract and introduction portion. Likewise, 

The credible sources of fossil energy efficiently are a vital cause of economic growth and considerable influence on adequate security. Whereas radiant energy positively enhances or ostensibly promotes socio-economic stability and the controlled environment. The fossil energy sources supply has become progressively stern in China and reconnoitering the beta decoupling relationships between CO2 emissions, GDP, energy consumption, electricity consumption, value-added industries, and population.

In the introduction portion, I have cited your research papers. As per the given valuable suggestion of your three research papers, I modified the 2nd paragraph lines (112-134) and included the valued finding of your research with technical skills. Furthermore, as per the given instruction research manuscript carefully check the theoretical. The whole paper screening again lines to the line under technical and academic writing style. And progressively improve the advised reader's economic interest. Graciously according to manuscript objectives, all the cleared tables, figures and values verify again with the proper interpretation and clear the determined objectives with technical skills. In addition, as per the given instruction, the theoretical check and modified the significant contribution from the line 253 to 265. 

Methods and Results

The results section of the paper presents a good view of the study. This work presents a notable investigation on a selected topic. I suggest including some graphical presentations to improve the quality of this study. Please see the proposed studies and see the graphical representation. Improve your work like these studies and cite them in this section.

Reply: Thank you very much for your valuable comments and suggestions. According to your valued comments, we have revised it carefully as follows. 

We have revised the paper point by point by the valuable comment.

As per your valuable suggestion, I have already included three figures (1,2 and 3) lines (287, 317, and 363) in the introduction portion in the form of primary energy consumption, electricity production, and value industries.

The second portion (methodology) included two figures (4 and 5) lines (502 and 525) in the form of the Beta decoupling and a strong positive/negative attitude of decoupling. The third portion (result and discussion) included three Figures (7,8 and 10) lines (720,807, and 958) in the form of GDP with Primary energy consumptions, energy generation, value-added primary industries, and Urban and rural population. CO2 emission industrial sectors (million tons) and the Pivot chart with GDP. In addition, individually interpreted all the eight figures in lines 269, 319, 365, 484, 508, 676,816, and 941. 

In this portion, I have also cited your three research papers and modified the separate sentences as for the valuable interest. 

Conclusion I suggest you make a separate heading of the conclusion and do not mix it with implications.  Policy Recommendations implication I again recommend you to make a separate heading of the Policy Recommendations.

The conclusion section is acceptable. Overall, this presents a good piece of research work. I recommend that authors do a little more work and revise this article accordingly. I suggest the authors check English quality and fix some weak sentences. If you have already taken English editing service, ask them to recheck the quality to meet scientific merit for publication. I endorse this manuscript for publication after minor corrections, as suggested.

Reply: Thank you very much for your valuable comments and suggestions. According to your valued comments, we have revised it carefully as follows. 

We have revised the paper point by point by the valuable comment.

As per the given instruction, I have separated both portion recommendation (line 1056) and implication also theoretical check it as per reader interest. 

Furthermore, the quality of English has been improved with editing services, and recheck the quality merit for publication.

Reviewer #3: Dear Authors, 

Thank you for the opportunity for me to go through and read your Paper Titled: Beta decoupling relationship between CO2 emissions by GDP, energy consumption, electricity production, value-added industries, and population in China.

The authors have done a great a job by revising the paper. The Abstract and Introduction has been revised Accordingly. Also, the Research Results is well explained and Demonstrated, therefore the recommend this paper for Publication.

Reply: Thank you very much for your valuable comments and suggestions. According to your valued comments, we have revised it carefully as follows. 

We have revised the paper point by point by the valuable comment.

Thank you so much for your kind compliments.

Thank you very much again for your valuable comments and suggestions.

---

## [Editor Report · Decision Letter 2]

19 Mar 2021

Beta decoupling relationship between CO2 emissions by GDP, energy consumption, electricity production, value-added industries, and population in China

PONE-D-20-34342R2

Dear Dr. Khan,

We’re pleased to inform you that your manuscript has been judged scientifically suitable for publication and will be formally accepted for publication once it meets all outstanding technical requirements.

Kind regards,

Bing Xue, Ph.D.

Academic Editor

PLOS ONE
---

## [Editor Report · Acceptance letter]

23 Mar 2021

PONE-D-20-34342R2 

Beta decoupling relationship between CO2 emissions by GDP, energy consumption, electricity production, value-added industries, and population in China 

Dear Dr. Khan:

I'm pleased to inform you that your manuscript has been deemed suitable for publication in PLOS ONE. Congratulations! Your manuscript is now with our production department. 

Kind regards, 

on behalf of

Professor Bing Xue 

Academic Editor

PLOS ONE